# The Impact of the COVID-19 Pandemic on the Tourism Sector in the Autonomous Region of Madeira

Paulo Rita *[ID], Nuno António [ID] and João Neves

NOVA Information Management School (NOVA IMS), Universidade NOVA de Lisboa, 1070-312 Lisbon, Portugal
* Correspondence: prita@novaims.unl.pt

**Abstract:** The highly regarded and award-winning tourism destination that is the autonomous region of Madeira (ARM), in the Madeira and Porto Santo islands, has suffered the consequences that the COVID-19 pandemic has brought to tourism through the mobility limitations as well as the fear faced by travelers. From data collected on tourism, COVID-19, and demography in ARM from the years 2019 to 2020, this study makes use of data science techniques, including statistics, data mining, and data visualization, to analyze the direct and indirect effects of the coronavirus outbreak as well as the weight of population density in the propagation of the virus. The results validate a direct effect and show evidence of dense regions having aggravated virus propagation, but they do not corroborate the idea that an indirect effect was significant.

**Keywords:** COVID-19; Madeira; tourism; population density; correlation significance analysis; k-means clustering

## 1. Introduction

The COVID-19 pandemic was a worldwide health emergency that tragically took the lives of over 2 million people globally [1]. To date, the transmission of this coronavirus is believed by scientists to be performed by direct transmission, from person to person, in close contact situations, or, on rare occasions, by touching contaminated surfaces or objects. In order to prevent the virus from spreading further, governments across the globe have followed the World Health Organization's (WHO) guidelines and implemented restrictive measures that range from schedule limitations and air traffic restrictions to complete lockdowns of entire regions.

Although these measures have shown effectiveness in stabilizing the spread of the virus, they dramatically offset the businesses forced to shut down their establishments and most businesses that relied on tourism for consumption. It has been previously determined that infectious disease outbreaks, including the coronavirus, greatly jeopardize the tourism industry, given its reliance on human mobility [2].

The tourism sector, being a highly vulnerable industry to various environmental, political, and socio-economic risks, is accustomed to and has become robust and resilient in recovering [3] from several distinct categories of crises, such as natural catastrophes, health emergencies, and terrorism attacks, among plenty others. Nevertheless, the nature and unprecedented events of the COVID-19 pandemic reveal signs that this health emergency is not only unique and unusual but has also produced profound and long-run transformations to the structure of tourism as an industry and its socio-economic activity [4,5].

Multiple authors have previously studied the impact of economic crises on tourism as well as the impact of pandemics on the economy. While not many have explored the impact of health-related crises on tourism, numerous new studies have emerged in recent times about the impact the COVID-19 pandemic has had on the tourism sector. Nevertheless, literature on the impact of pandemics on tourism has uncovered a number of gaps. There appears to be a controversy in the results regarding whether population density has a



negative impact on the propagation and severity of pandemics. Concerning COVID-19, some authors, namely [6–8], refer to population level indicators as highly and significantly correlated to the number of infections. Thus, this indicator is considered a determinant in the proliferation of viruses, and other authors have obtained results that do not corroborate the idea that population density is a determining factor of influence in the context of the pandemic. A study by [9] recalls the case of some extremely populationally dense cities, such as Singapore, Seoul, and Shanghai, that have outperformed other less dense cities in fighting the COVID-19 pandemic and uses an empirical approach to study the impact of the population density of Chinese cities on the proliferation of the coronavirus, and its results found no significant correlation between the two either. The opposing results among the authors do not allow for a clear answer to the given question, therefore originating a gap in the research where there is no objective conclusion.

Furthermore, the literature that studies the indirect impact of pandemics on tourism tends to use a theoretical approach. Of the few studies that followed an empirical approach, most used questionnaires to assess the population's opinion, demonstrating a lack of author contributions and real daily data. The use of real data are recognized to have higher reliability and accuracy than questionnaire data [10].

Considering the identified gaps, this study aims to assess the direct and indirect impact of the pandemic on the tourism sector in the autonomous region of Madeira. This study also aims to understand whether the impact of the pandemic varied by municipality. This type of study is necessary to reinforce the knowledge available for governmental decision-making for future pandemics or health-related crises based on cyclical patterns, which tend to recur with various types of viruses and bacteria, reappearing inevitably at some point in time.

As Portugal's third most visited region, following the Algarve and Lisbon, the autonomous region of Madeira (ARM) is an enticing archipelago renowned for its lush landscapes, unique flora and fauna, vibrant culture, and the world-famous Madeira wine [11]. Historically, the autonomous region of Madeira is notorious for having a local economy heavily reliant on tourism activity as its primary source of income, with 26% of its regional GDP associated with tourism products. The sector is responsible for 20 thousand workstations in the region, and therefore, it is no surprise that the archipelago was deeply affected by the pandemic. In 2020, the tourism sector was virtually disabled for several months and heavily restricted upon reopening, and as such, the major source of income for the local economy was cut down, and the whole sector was drastically impacted. Statistics published by DREM show that the sector was reduced to zero activity every month, and the region's GDP was reduced by 2.2%. According to the same source, this sector's indicators, such as the arrivals and sleepovers of tourists in the region, the income of accommodation services, and occupation rates of these services, among others, dropped dramatically to zero or extremely low values in the second trimester of 2020, leaving it clear that the damages caused by the pandemic were devastating not only for the industry and healthcare sectors but also to the tourism sector [12].

In particular, this study tries to answer four research questions. The first research question is whether or not there was a direct impact of the COVID-19 pandemic on tourism in the year 2020, in the autonomous region of Madeira, in light of what is suggested by the vast majority of the authors [13,14] as well as [15] and in domestic studies on the impacts of the COVID-19 crisis on the tourism expectations of the Azores Archipelago residents [16]. Although most studies support this direct impact and its expected results are similar to those obtained by the authors mentioned above, this is seen as a highly relevant subject of analysis in the context of the topic of this study.

Similarly, the second research question is whether or not there was an indirect impact of the COVID-19 pandemic on tourism, in light of studies such as the one made by [17] on the indirect effect of malaria outbreaks on tourism in African regions where there were no cases and also in a very recent study that focuses on how fear aggravated the damage caused to tourism in China by the COVID-19 pandemic, as suggested by [18].

The third research question is whether the population density of the municipalities significantly influenced the spreading of the virus in the autonomous region of Madeira, and it originates from several research papers that have focused their studies on the link between the population density of the territories and the current pandemic. As mentioned in the literature review, the results among studies have been somewhat contradictory, but the results seem to converge to a positive relationship [6–8,19–26]. However, other studies point to the fact that the density of the population is not significant and cannot be a determining factor in the spread of COVID-19 [9]. The study by [27] evaluates this effect as non-significant in earlier stages, with increasing significance in later stages of the pandemic.

The fourth research question is whether there are chunks of similar data that can be grouped to identify and classify affected municipalities, as suggested by the studies above [7], or if the region handled the virus homogeneously.

## 2. Literature Review

Epidemic and pandemic emergencies often provoke critical negative swings in demand for usually popular travel destinations, as tourists may, knowing of the risks, cancel their trips in case they opt not to expose themselves to such dangers, becoming contaminated or even restrained in a foreign location indeterminably [28]. Each person's perceived risk associated with traveling during outbreaks has been shown to affect their willingness to travel [29]. A study published by representatives of the University of Technology Republic of South Africa [17] focuses on the indirect impact of pandemics on tourism in Africa. The study addresses the example of the Ebola crisis, which affected various African tourism destinations, which experienced lower travel demand and, therefore, lower tourism consumption, some of which was due to direct consequences of the pandemic. However, it also refers to the fact that the travelers canceled their trips even to faraway lands such as South Africa, Kenya, and West Africa, with no reported virus cases.

Furthermore, hotel occupancy rates in Nigeria dropped by half due to media coverage of Ebola before cases were reported in the region. The authors additionally address the Zika outbreak in South America, which was declared a public health emergency by the World Health Organization (WHO), and the declaration itself showed a negative impact on sport tourism gatherings and religious gatherings. The study focuses on the indirect effect of these health emergencies on tourism without directly affecting a region. On a similar note, a recent study focuses on the COVID-19 pandemic and how it generated an unprecedented level of public fear, and it studies how such fear aggravated the damage caused to tourism in China during the pandemic [18]. Other studies analyzed the impact of the fear caused by the SARS virus outbreak in 2004 and how it reduced people's propensity to travel. The consensus amongst the authors is that multiple traveling fear-inducing factors emerge from health emergencies, and even in regions that have not suffered directly from them, tourism has indirect consequences that originate from the effects of the media, international tourism, and globalization [30].

In a chain of events, factors that promote the spread of diseases inevitably end up damaging the tourism sector by amplifying the dimension taken by the outbreaks. Multiple authors have highlighted that population density is a major source of concern for health and governmental authorities, especially in the case of highly contagious diseases such as the one caused by the novel SARS-CoV-2 virus, which is the reason why experts have claimed physical distancing to be one of the most effective measures to fight the spread of the virus [27]. Regarding the available literature on the relationship between population density and the spread of pandemics, there has been a lack of historical documentation. Highlighted in the context of this relationship are several studies [22–24,26], which, despite using different approaches and methodologies, have shown converging results regarding this relationship, which are that of a strong influence of population density on the rate of spread of pandemics and epidemics and indicate a positive relationship between this indicator and the speed at which diseases spread out. However, in recent times, several research

papers have been studying the link between the population density of the territories and the current COVID-19 pandemic, and the results among the studies have been somewhat contradictory, though the relation is positive the majority of the time. Ref. [21] studied how this relationship panned out in the United States, particularly in the state of Alabama, and found that, despite having less testing per population density, new infections were disproportionately more frequent in heavily populated regions, indicating that not only infections were more prominent in highly dense regions of the state but also that instances in these areas may even be underreported. Ref. [25] dove deeper into the United States case by studying the relationship between this indicator and the reproductive number of infections across the country's counties, verified with a sensitivity analysis of the results. Their findings were that this relationship is positive and significant across counties, regardless of multiple other factors, such as public transportation usage versus personal vehicle ownership and household income. The relationship is possibly justified due to higher contact rates due to higher population density. Conversely, in the geographical context of the United States, a bivariate and multivariate regression approach has been applied to indicate that population density has little significance when explaining the number of infections throughout the country. However, it became a good predictor of the results of cumulative infections as the virus spread across the United States, concluding that this indicator, while not being as good of a predictor in the early stages of a pandemic, has shown its weight as the infectious disease started spreading and reached later stages [27].

When looking at the case of India, one of the densest countries in the world, studies made on the influence of population density and the COVID-19 pandemic in Indian states have shown that, even in states with the most sophisticated healthcare infrastructure, the spatial analysis has shown that density strongly influenced the virus' transmission rate [19]. A similar study by [31] also analyzes how population density has impacted infection and mortality rates in Indian cities by using Pearson R and regression, and its results also indicate that the relationship is positive. Ref. [20] also uses a regression model to investigate how this relationship performs in Turkish cities by measuring the impact of population density on the elasticity of the curve that is drawn by infection cases and finds that density accentuates the rate at which the cases rise and that lower densities are linked to values of elasticity that are close to or even lower than 1, which means that the curve tends to flatten out for the lowest density cities. On the same note, the results show that the curve rises most of the time and becomes steeper as density increases. Ref. [32] adds to this topic by analyzing how the virus spread was influenced by wind and population density in 81 provinces in Turkey and also finding that dense provinces had a faster rate of infections. Equally, dense provinces were negatively affected by wind, assuming that higher wind speeds increase air circulation and promote transmission. The two parameters were found to explain 94% of the variance of virus spreading and thus were concluded to have had a significant influence on the proliferation of the virus, particularly when working together.

The French study by [6] suggested that the link between the density of French territories based on data from the 2016 Census and the epidemic was positive most of the time, assuming that a higher density would result in a higher propensity for one to become contaminated or by a higher death rate, despite Chinese studies stating otherwise, leveraging the distance to Wuhan, the epicenter of the epidemic, as a bigger factor than population density regarding this relationship in China. The study by [7] focuses on population density as a factor in the spread of COVID-19 in Algeria, where a clustering algorithm allowed to isolate the groups of cities with higher numbers of COVID-19 infections as well as the highest population densities and found strong correlations associated with high significance regarding that relationship. The data analysis findings verified that population density positively affected the spread of COVID-19 in Algeria. Additionally, a study of European countries and the USA determined that population density has a small but substantial effect on the rate of spread of the virus and claims that there is a significant correlation between these two variables with a correlation coefficient of $R2 = 0.23$ in Europe and $R2 = 0.39$ in the USA [8]. Conversely, other analyses believe that the density of the popu-

lation is not a topic of concern and cannot be a determining factor in the proliferation of pandemics such as COVID-19 and use cities such as Singapore, Seoul, Shanghai, and New York as counterexamples because of their underlying dense populations, whose number of infections caused by the COVID-19 pandemic was no different from the ones documented in cities with low urban density in per capita terms.

Similarly, in an empirical study from China on data collected from 284 Chinese cities, the results do not corroborate the idea that population density is a determining factor for the transmission of COVID-19. On the other hand, the most afflicted cities have a relatively low density between 5000 and 10,000 inhabitants per $km^2$ [9]. Other studies have also found other inconsistent results regarding this relationship, showing no significant relationship between COVID-19 spreading and population density, and explain it through the heavy, restrictive measures used by the Chinese government, which were one of the strictest across the globe, which effectively reduced the extent to which density could affect a country that was on an entire lockdown and heavily restricted human-to-human contact.

Several authors have also contributed to the topic of the negative direct impact of the pandemic on the tourism sector. The first of the studies is the one by [13], which focuses on the direct impact of COVID-19 on the tourism industry in Malaysia, particularly on the airlines and hotel businesses, having reached conclusive results on the dramatic damage caused to the sector due to, not only the increase in cases but also revealing a large number of tourists that canceled their trips due to the Malaysian government imposing travel restrictions and bans, most of which were recommended or even imposed by the World Health Organization. There is also the study by [14] that also focuses on how COVID-19 induced a global change and hindered tourism worldwide and how they compared with previous pandemics, concluding that the COVID-19 pandemic had a magnitude in tourism that was never seen before, which also revealed and raised questions about the vulnerability of work posts, particularly low-wage work posts in the tourism sector, which were disproportionately affected by the crisis, especially in lower-income countries, exposing this weakness in the sector that is conceivable to be affected similarly by future health crises. Furthermore, the study by [15] also emphasizes the quarantine's impact on the tourism industry in Lviv, Ukraine, due to COVID-19. Its results show that the pandemic had a massive negative effect on all of the indicators of the tourism sector in Lviv in 2020, with a loss of tourism flow, expense, and budget, among others, and a severe economic and market crisis associated with it. In the domestic panorama, we have studied [16] the impacts of the COVID-19 crisis on the tourism expectations of the Azores Archipelago residents, and the results have shown that most of the residents had their travel expectations significantly lowered directly due to the pandemic. However, the results might include a significant level of indirect influence from the pandemic on the survey answers from the study.

Lastly, the study by [33] analyzes the COVID-19 pandemic through cluster analysis as a data mining process, finding groups of states with similar reactions to the pandemic regarding cured and death cases. This study would be interesting to see a more in-depth look at the demographics and other data and evaluate whether other variables might have influenced some countries to behave differently from others and which variables make some countries behave alike.

## 3. Methodology

The methodology chosen is the cross-industry standard process for data mining (CRISP-DM), a data mining model that uses the best practices to explore and analyze data. This model has been traditionally broken down into six steps: Business Understanding, Data Understanding, Data Preparation, Modeling, Evaluation, and Deployment.

### 3.1. Business Understanding

In this phase, the key is to determine the objectives of the data mining project. This phase involves identifying the available assets and resources, their associated constraints, and the objectives to be achieved with the project.

For this study, the objectives of the data mining project rely on gathering data and information about the tourism sector and the COVID-19 pandemic and using descriptive data mining tools to transform this data into knowledge, draw valuable conclusions, and extract meaningful results from previously fragmented data.

The data was mainly retrieved from the Direção de Regional de Estatística da Madeira (DREM) Tourism Reports from 2019 to 2020, the DREM Demographic Report from 2020, the Yearly Reports from Madeira Ports Association (APRAM) from 2019 to 2020 with data from cruise and merchandise ship movement, and COVID-19 data from Direção Geral de Saúde (DGS). The data collected were scattered across 16 tables and involved the 33 variables in Table 1.

**Table 1.** Variables' Metadata.

| Variable | Meaning |
| --- | --- |
| newguests_2019 | New guests arriving in hotels in 2019 |
| newguests_2020 | New guests arriving in hotels in 2020 |
| newguests_var | Variation of new guests arriving in hotels 2019–2020 |
| guests_2019 | Lodged guests in 2019 |
| guests_2020 | Lodged guests in 2020 |
| guests_var | Variation of lodged guests 2019–2020 |
| sleepovers_2019 | Sleepovers in hotels in 2019 |
| sleepovers_2020 | Sleepovers in hotels in 2019 |
| sleepovers_var | Variation of sleepovers in hotels 2019–2020 |
| totalincome_2019 | Total hotel income 2019 |
| totalincome_2020 | Total hotel income 2020 |
| totalincome_var | Variation in hotel income 2019–2020 |
| totalroomincome_2019 | Total room income 2019 |
| totalroomincome_2020 | Total room income 2020 |
| totalroomincome_var | Variation in total room income 2019–2020 |
| personelcosts_2019 | Hotel personnel cost in 2019 |
| personelcosts_2020 | Hotel personnel cost in 2020 |
| personelcosts_var | Variation in hotel personnel costs 2019–2020 |
| stopovers_2019 | Ships stopping over at the shore 2019 |
| embarked_2019 | Total of ship embarkments in 2019 |
| disembarked_2019 | Total of ship disembarkments in 2019 |
| intransit_2019 | Total ships in transit in 2019 |
| stopovers_2020 | Ships stopping over at the shore 2020 |
| embarked_2020 | Ships stopping over at the shore 2020 |
| disembarked_2020 | Total of ship disembarkments in 2020 |
| intransit_2020 | Total ships in transit in 2020 |
| covid19casesportugal_2020 | Registered COVID-19 Cases in Portugal |
| covid19casesmadeira_2020/covidcases | Registered COVID-19 Cases in Madeira |
| averagestay | Average time spent lodged in the region (days) |
| populationaldensity | Number of inhabitants per km$^2$ |
| longevity | Measure of population life expectancy |
| avgpopulation | Average population |
| worldcases | Registered COVID-19 cases worldwide |

The software used for the data pre-processing was Python, using the Anaconda Notebook. The variables already include some calculated variables, such as the yearly percent variation of the arrival of tourists.

The 33 variables are described using two different approaches, scrutinized by month or municipality. In order to analyze this data, the 16 original tables were merged into only 2, which group data of each type, the Monthly Report and the Municipality Report:

Monthly Report: The data are scrutinized by month. The records are from January to December, and the variables include most of the earlier mentioned tourism indicators, COVID-19 variables, and demographic indicators of the whole Archipelago.

Municipality Report: The data are scrutinized by municipality. The records are the 11 municipalities of the Archipelago—Funchal, Machico, Ribeira Brava, Santa Cruz, Câmara

de Lobos, Calheta, Ponta do Sol, São Vicente, Porto Moniz, Santana, and Porto Santo—and again, the variables include most of the tourism indicators mentioned above, COVID-19 variables, and demographic indicators.

### 3.2. Data Understanding

The data is collected and explored during this phase to understand its content, shape, structure, and properties. This is also the phase where the appropriate statistical tools and algorithms are determined to be more appropriate during the modeling phase.

For this study, data was collected from the sources previously mentioned. An overview was made of what variables could be extracted directly from the data sources, what other variables were needed, and if they could be created from existing ones, as well as identifying purposeless or redundant variables for the study to be discarded and creating preliminary graphs and charts to visualize the raw, initial data that allowed a personal interpretation and understanding of the available resources. In this phase, the main tools used were JMP and Python. Within Python, packages and libraries such as Pandas, Numpy, and Matplotlib were used, among plenty of others, to perform preliminary adjustments, column filtering, and evaluating data coherence.

### 3.3. Data Preparation

This phase includes cleaning, validation, remapping, and transformations of data, and the tools used were a combination of Python 3.10, Excel 2016, and JMP 16 by SAS. For this study, due to the nature of the data obtained from official institutions or governmental establishments, data cleaning was conducted only as a methodology principle. The validation process detected scarce missing, null, or duplicate values and inconsistencies. However, several transformations, including merging and remapping of data, were required since the elements were scattered across several sources and tables, so the procedure began by reorganizing the resources into new tables. By the end of this phase, the final datasets were ready to be used in the following modeling phase.

### 3.4. Modeling

After the data were remapped and ready to be analyzed, the first step was an exploratory data analysis, which began with a univariate analysis to study the variables' variances and outliers and determine whether normalization is required. Secondly, a bivariate analysis was also conducted, with a Pearson correlation analysis, where multi-collinearity among the variables was checked.

If the correlation coefficient ranges between $r = 0.50$ and $r = 1.00$, it depicts a strong positive or high degree of relationship between the two variables. If the correlation coefficient ranges between $r = -1.00$ and $r = -0.50$, it relates to a strong negative relationship. If the correlation coefficient approaches $r = 0.00$, it indicates no correlation between the two variables.

Specific models and algorithms are selected and run on the data during the modeling phase. First, it is crucial to dive deeper than correlation analysis when dealing with multivariate data since it is never appropriate to conclude that changes in one variable cause changes in another based only on correlation alone, especially when dealing with subsets of data. Therefore, following the correlation analysis, a significance testing analysis was also performed to determine whether the relationships between variables were causal or meaningful by computing their statistical significance, which is obtained by evaluating the linear relationship between them. For this study, the method used in this stage was the *p*-value method, whose *p*-value score was then put under scrutiny by performing statistical testing to determine whether the correlation coefficient was significant.

Therefore, after setting $\alpha$, the significance level, in the case where:

*p*-value $\leq \alpha$: The correlation is considered statistically significant.
*p*-value $> \alpha$: The correlation is not considered statistically significant.

Furthermore, a clustering analysis was made to group the municipalities according to their associated variables in the Municipality Report to create segments of the study of the

composition of each cluster and, in particular, with interest in studying how COVID-19 infections and population densities behave among clusters. The algorithm used was k-means, an unsupervised algorithm that, given a k number of nearest neighbors, partitions n observations into k clusters, in which each observation is assigned to the cluster with the nearest mean, or centroid, minimizing the Euclidean distance of the observations concerning the centroid, which begin at given points for every cluster, and then as the algorithm runs in an iterative process, calculations are made to optimize the positions of the centroids, resulting in the formation of k clusters.

### 3.5. Evaluation

In this phase of the project, a review of the models is made to determine their accuracy and ability to meet the goals and objectives of the project identified in previous phases. So, in this case, the goal is to answer the questions asked in the model, extract conclusions about the results, and study similarities and differences in the clustering models.

### 3.6. Deployment

Finally, the deployment phase includes disseminating the information, which includes the tables and dashboards created within the tools used. In this case, these include the correlation and significance tables of all the tables used and the results of the clustering algorithm, complemented by the necessary reports to support them.

## 4. Results and Discussion

Correlations were calculated for every pair of variables in both datasets. Data analysis was made to test the significance of these relationships. We wanted to study if there was a relationship between worldwide COVID-19 infections and the arrival of new guests at lodging services in 2020 and if that would be negative. However, the results do not corroborate this idea. In the first half of the pandemic, we can observe that this relationship was linear and positive, after which it became irregular, as shown in Figures 1 and 2.

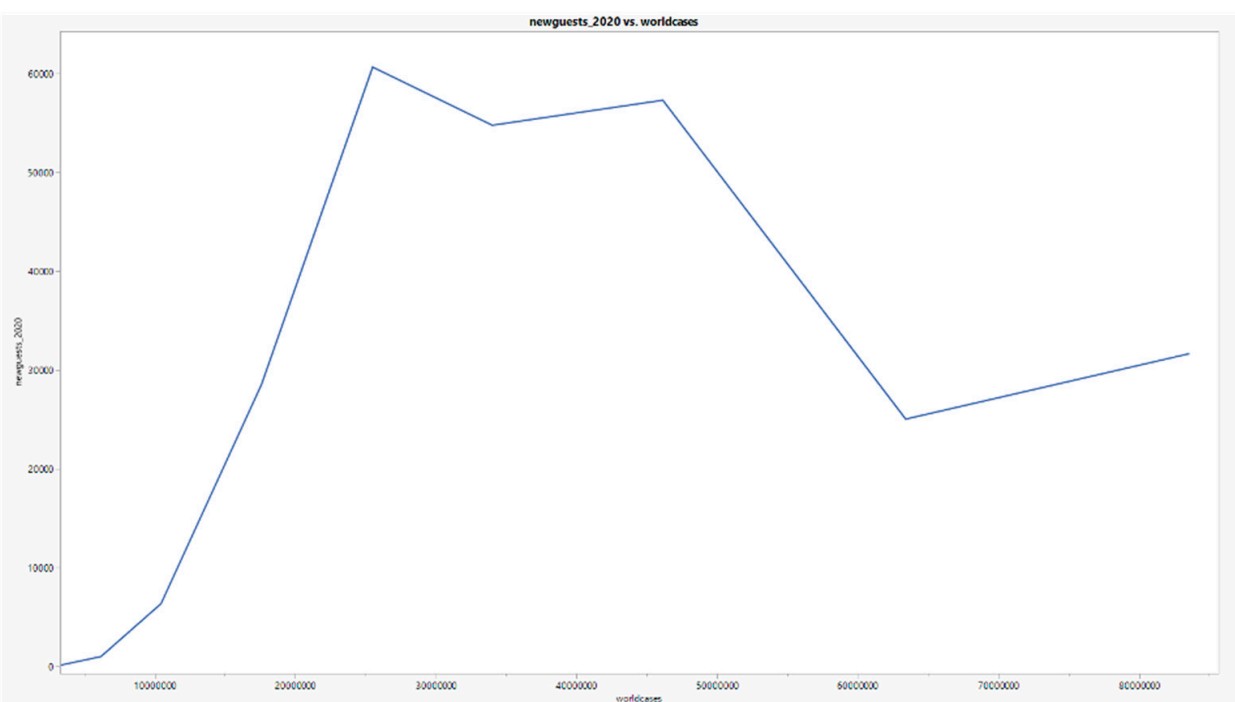

**Figure 1.** Evolution of new lodged guests per worldwide infection count.

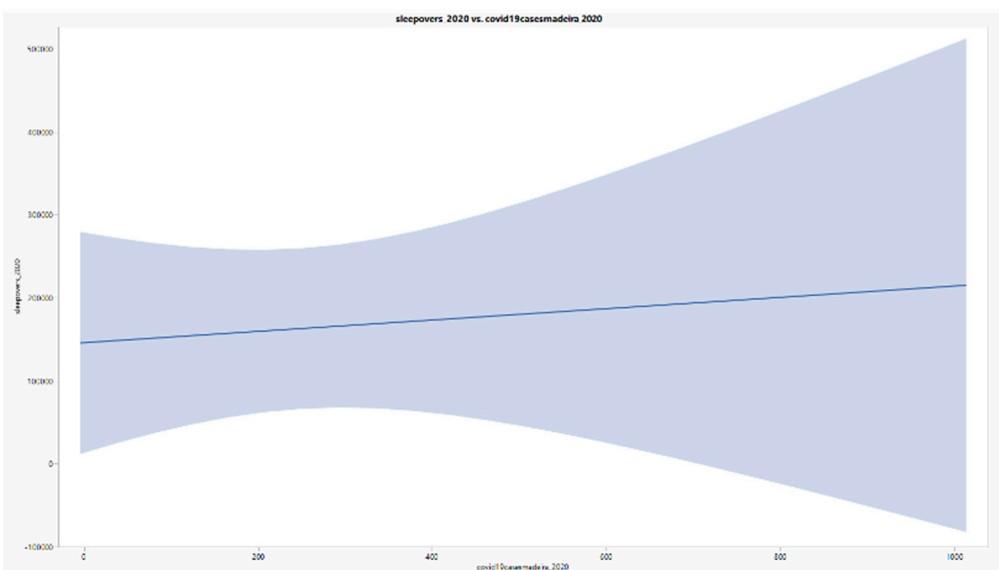

**Figure 2.** Line of Fit. Sleepover variation by regional infection count.

Firstly, a significance level of $\alpha$ was set at $\alpha = 0.05$ to evaluate the statistical significance of the correlation between registered COVID-19 cases in the region and the tourism variables. Highlighted in Figures 4 and 6 are the *p*-values of such correlations, referring to the Monthly Report and the Municipality Report, respectively. Also highlighted in red are the results whose *p*-value is smaller than the significance level $\alpha$. This means there is sufficient evidence to conclude that there is a significant linear relationship between those variables and the count of infections due to the correlation coefficient being significantly different from zero. If a slightly less usual significance level of $\alpha = 0.1$ were to be assumed, variables "Longevity" and "COVID-19 cases in Madeira" would also become significantly correlated, which is an interesting result since older age groups, and in particular rural populations, tend to overlook and neglect these sorts of health crises, possibly explaining this result. However, keeping the current $\alpha$, this correlation remains statistically insignificant.

Analyzing the significance of the variables' "New guests", "Guests", "Sleepovers" and "Average Stay" correlation with "COVID-19 cases in Madeira", the results might appear counterintuitive at first since they are positively correlated. However, one can consider these values to reflect the region's governmental policies. Such policies include testing the travelers arriving in ARM and the mandatory hotel quarantine until they receive their test results. Thus, these policies can explain why an increase in cases leads to more people being lodged in tourism accommodation services and consequently explain the correlation and significance level between the variables.

The correlation matrix for monthly data (Figure 3) shows that most of the correlations that made sense to investigate seem to have counterintuitive values and are not objectively what was expected. Additionally, by analyzing the values obtained in the *p*-value matrix (Figure 4), the results show that, when looking at the confirmed cases in the region, there is a pattern of negative correlations with most of the remaining variables. These results point towards the negative relationship that was expected. However, the *p*-value analysis of these correlations has revealed no statistical significance in the correlations between COVID-19 cases and the tourism indicators from 2020, so the results do not validate the significance of the negative relationship found and are thus not in line with the results obtained by [13–16], where this relationship was found to be positive.

| | newguests_2020 | newguests_var | guests_2020 | guests_var | sleepovers_2020 | sleepovers_var | totalincome_2020 | totalincome_var | totalroomincome_2020 |
|---|---|---|---|---|---|---|---|---|---|
| newguests 2020 | 1 | 0.9501 | 0.9928 | 0.943 | 0.9778 | 0.9201 | 0.9753 | 0.9122 | 0.9765 |
| newguests_var | 0.9501 | 1 | 0.9648 | 0.9974 | 0.9828 | 0.9919 | 0.9789 | 0.9888 | 0.9796 |
| guests_2020 | 0.9928 | 0.9648 | 1 | 0.9648 | 0.9933 | 0.9461 | 0.9923 | 0.9401 | 0.9929 |
| guests_var | 0.943 | 0.9974 | 0.9648 | 1 | 0.9866 | 0.997 | 0.9827 | 0.9951 | 0.9829 |
| sleepovers_2020 | 0.9778 | 0.9828 | 0.9933 | 0.9866 | 1 | 0.9755 | 0.9978 | 0.9714 | 0.9982 |
| sleepovers_var | 0.9201 | 0.9919 | 0.9461 | 0.997 | 0.9755 | 1 | 0.9717 | 0.9997 | 0.9715 |
| total income_2020 | 0.9753 | 0.9789 | 0.9923 | 0.9827 | 0.9978 | 0.9717 | 1 | 0.9678 | 0.9999 |
| totalincome_var | 0.9122 | 0.9888 | 0.9401 | 0.9951 | 0.9714 | 0.9997 | 0.9678 | 1 | 0.9674 |
| totalroomincome_2020 | 0.9765 | 0.9796 | 0.9929 | 0.9829 | 0.9982 | 0.9715 | 0.9999 | 0.9674 | 1 |
| totalroomincome_var | 0.9123 | 0.9888 | 0.9385 | 0.9944 | 0.9699 | 0.9994 | 0.9658 | 0.9998 | 0.9655 |
| personelcosts_2020 | 0.8652 | 0.8529 | 0.8888 | 0.8652 | 0.8819 | 0.8525 | 0.8994 | 0.8506 | 0.8964 |
| personelcosts_var | 0.8962 | 0.8855 | 0.9242 | 0.9004 | 0.9207 | 0.9353 | 0.9353 | 0.8873 | 0.9325 |
| stopovers_2020 | 0.6711 | 0.8394 | 0.7381 | 0.8584 | 0.8004 | 0.873 | 0.7908 | 0.8771 | 0.7923 |
| embarked_2020 | 0.7281 | 0.8241 | 0.7757 | 0.8515 | 0.8266 | 0.8761 | 0.8128 | 0.882 | 0.8127 |
| disembarked_2020 | 0.6979 | 0.7989 | 0.7605 | 0.8336 | 0.8118 | 0.8571 | 0.8043 | 0.8639 | 0.8038 |
| Intransit_2020 | 0.6969 | 0.8598 | 0.7573 | 0.8769 | 0.8202 | 0.8935 | 0.8076 | 0.8972 | 0.8092 |
| covid19casesportugal_2020 | -0.2007 | -0.1799 | -0.2039 | -0.1793 | -0.2156 | -0.1778 | -0.2066 | -0.1718 | -0.2121 |
| covid19casesmadeira_2020 | -0.1508 | -0.153 | -0.1876 | -0.176 | -0.2041 | -0.1701 | -0.1587 | -0.1677 | -0.163 |

| | totalroomincome_var | personelcosts_2020 | personelcosts_var | stopovers_2020 | embarked_2020 | disembarked_2020 | intransit_2020 | covid19casesportugal_2020 | covid19casesmadeira_2020 |
|---|---|---|---|---|---|---|---|---|---|
| newguests_2020 | 0.9123 | 0.8652 | 0.8962 | 0.6711 | 0.7281 | 0.6979 | 0.6969 | -0.2007 | -0.1508 |
| newguests_var | 0.9588 | 0.8529 | 0.8855 | 0.8394 | 0.8241 | 0.7989 | 0.8598 | -0.1799 | -0.153 |
| guests_2020 | 0.9385 | 0.8888 | 0.9242 | 0.7381 | 0.7757 | 0.7605 | 0.7573 | -0.2039 | -0.1876 |
| guests_var | 0.9944 | 0.8652 | 0.9004 | 0.8584 | 0.8515 | 0.8336 | 0.8769 | -0.1793 | -0.176 |
| sleepovers_2020 | 0.9698 | 0.8819 | 0.9207 | 0.0004 | 0.8266 | 0.8118 | 0.8202 | -0.2156 | -0.2041 |
| sleepovers_var | 0.9994 | 0.8525 | 0.8889 | 0.873 | 0.8761 | 0.8571 | 0.8935 | -0.1778 | -0.1701 |
| totalincome_2020 | 0.9658 | 0.8994 | 0.9353 | 0.7908 | 0.8128 | 0.8043 | 0.8076 | -0.2066 | -0.1587 |
| totalincome_var | 0.9998 | 0.8506 | 0.8873 | 0.8771 | 0.882 | 0.8639 | 0.8972 | -0.1718 | -0.1677 |
| totalroomincome_2020 | 0.9655 | 0.8964 | 0.9325 | 0.7923 | 0.8127 | 0.8038 | 0.8092 | -0.2121 | -0.163 |
| totalroomincome_var | 1 | 0.8482 | 0.884 | 0.8708 | 0.882 | 0.8607 | 0.8926 | -0.1658 | -0.1605 |
| personelcosts_2020 | 0.8482 | 1 | 0.9918 | 0.6386 | 0.6728 | 0.6924 | 0.6385 | 0.0643 | 0.0666 |
| personelcosts_var | 0.884 | 0.9918 | 1 | 0.6952 | 0.7373 | 0.7582 | 0.6977 | -0.0146 | -0.0294 |
| stopovers_2020 | 0.8708 | 0.6356 | 0.6952 | 1 | 0.8551 | 0.8837 | 0.9946 | -0.3055 | -0.3892 |
| embarked_2020 | 0.882 | 0.6728 | 0.7373 | 0.8551 | 1 | 0.9851 | 0.8926 | -0.3535 | -0.4228 |
| disembarked_2020 | 0.8607 | 0.6924 | 0.7582 | 0.8837 | 0.9851 | 1 | 0.9062 | -0.3598 | -0.4309 |
| intransit_2020 | 0.8926 | 0.6385 | 0.6977 | 0.9946 | 0.8926 | 0.9062 | 1 | -0.3372 | -0.4025 |
| covid19casesportugal_2020 | -0.1658 | 0.0643 | -0.0146 | -0.3055 | -0.3535 | -0.3598 | -0.3372 | 1 | 0.5069 |
| covid19casesmadeira_2020 | -0.1605 | 0.0666 | -0.0294 | -0.3892 | -0.4228 | -0.4309 | -0.4025 | 0.5069 | 1 |

**Figure 3.** Correlation Matrix—Monthly Data.

| | newguests_2020 | newguests_var | guests_2020 | guests_var | sleepovers_2020 | sleepovers_var | totalincome_2020 | totalincome_var | totalroomincome_2020 |
|---|---|---|---|---|---|---|---|---|---|
| newguests_2020 | <0.0001 | <0.0001 | <0.0001 | <0.0001 | <0.0001 | <0.0001 | <0.0001 | <0.0001 | <0.0001 |
| newguests_var | <0.0001 | <0.0001 | <0.0001 | <0.0001 | <0.0001 | <0.0001 | <0.0001 | <0.0001 | <0.0001 |
| guests_2020 | <0.0001 | <0.0001 | <0.0001 | <0.0001 | <0.0001 | <0.0001 | <0.0001 | <0.0001 | <0.0001 |
| guests_var | <0.0001 | <0.0001 | <0.0001 | <0.0001 | <0.0001 | <0.0001 | <0.0001 | <0.0001 | <0.0001 |
| sleepovers_2020 | <0.0001 | <0.0001 | <0.0001 | <0.0001 | <0.0001 | <0.0001 | <0.0001 | <0.0001 | <0.0001 |
| sleepovers_var | <0.0001 | <0.0001 | <0.0001 | <0.0001 | <0.0001 | <0.0001 | <0.0001 | <0.0001 | <0.0001 |
| total income_2020 | <0.0001 | <0.0001 | <0.0001 | <0.0001 | <0.0001 | <0.0001 | <0.0001 | <0.0001 | <0.0001 |
| totalincome_var | <0.0001 | <0.0001 | <0.0001 | <0.0001 | <0.0001 | <0.0001 | <0.0001 | <0.0001 | <0.0001 |
| totalroomincome_2020 | <0.0001 | <0.0001 | <0.0001 | <0.0001 | <0.0001 | <0.0001 | <0.0001 | <0.0001 | <0.0001 |
| totalroomincome_var | <0.0001 | <0.0001 | <0.0001 | <0.0001 | <0.0001 | <0.0001 | <0.0001 | <0.0001 | <0.0001 |
| personelcosts_2020 | 0.0003 | 0.0004 | 0.0001 | 0.0003 | <0.0001 | 0.0004 | <0.0001 | 0.0005 | <0.0001 |
| personelcosts_var | <0.0001 | 0.0001 | <0.0001 | <0.0001 | <0.0001 | 0.0001 | <0.0001 | 0.0001 | <0.0001 |
| stopovers_2020 | 0.0169 | 0.0006 | 0.0061 | 0.0004 | 0.0018 | 0.0002 | 0.0022 | 0.0002 | <0.0001 |
| embarked_2020 | 0.0072 | 0.001 | 0.003 | 0.0004 | 0.0009 | 0.0002 | 0.0013 | 0.0001 | 0.0021 |
| disembarked_2020 | 0.0116 | 0.0018 | 0.0041 | 0.0008 | 0.0013 | 0.0004 | 0.0016 | 0.0003 | 0.0016 |
| Intransit_2020 | 0.0118 | 0.0003 | 0.0043 | 0.0002 | 0.0011 | <0.0001 | 0.0015 | <0.0001 | 0.0014 |
| covid19casesportugal_2020 | 0.5317 | 0.5758 | 0.5251 | 0.5772 | 0.501 | 0.5803 | 0.5194 | 0.5933 | 0.5082 |
| covid19casesmadeira_2020 | 0.64 | 0.6351 | 0.5592 | 0.5843 | 0.5246 | 0.5972 | 0.6222 | 0.6024 | 0.6127 |

| | totalroomincome_var | personelcosts_2020 | personelcosts_var | stopovers_2020 | embarked_2020 | disembarked_2020 | intransit_2020 | covid19casesportugal_2020 | covid19casesmadeira_2020 |
|---|---|---|---|---|---|---|---|---|---|
| newguests_2020 | <0.0001 | 0.0003 | <0.0001 | 0.0169 | 0.0072 | 0.0116 | 0.0118 | 0.5317 | 0.64 |
| newguests_var | <0.0001 | 0.0004 | 0.0001 | 0.0006 | 0.001 | 0.0018 | 0.0003 | 0.5758 | 0.6351 |
| guests_2020 | <0.0001 | 0.0001 | <0.0001 | 0.0061 | 0.003 | 0.0041 | 0.0043 | 0.5251 | 0.5592 |
| guests_var | <0.0001 | 0.0003 | <0.0001 | 0.0004 | 0.0004 | 0.0008 | 0.0002 | 0.5772 | 0.5843 |
| sleepovers_2020 | <0.0001 | 0.0001 | <0.0001 | 0.0018 | 0.0009 | 0.0013 | 0.0011 | 0.501 | 0.5246 |
| sleepovers_var | <0.0001 | 0.0004 | 0.0001 | 0.0002 | 0.0002 | 0.0004 | <0.0001 | 0.5803 | 0.5972 |
| totalincome_2020 | <0.0001 | <0.0001 | <0.0001 | 0.0022 | 0.0013 | 0.0016 | 0.0015 | 0.5194 | 0.6222 |
| totalincome_var | <0.0001 | 0.0005 | 0.0001 | 0.0002 | 0.0001 | 0.0003 | <0.0001 | 0.5933 | 0.6024 |
| totalroomincome_2020 | <0.0001 | <0.0001 | <0.0001 | 0.0021 | 0.0013 | 0.0016 | 0.0014 | 0.5082 | 0.6127 |
| totalroomincome_vari | <0.0001 | 0.0005 | 0.0001 | 0.0002 | 0.0001 | 0.0003 | <0.0001 | 0.6065 | 0.6182 |
| personelcosts_2020 | 0.0005 | <0.0001 | <0.0001 | 0.0254 | 0.0165 | 0.0126 | 0.0255 | 0.7945 | 0.8371 |
| personelcosts_var | 0.0001 | <0.0001 | <0.0001 | 0.0121 | 0.0062 | 0.0043 | 0.0116 | 0.9641 | 0.9277 |
| stopovers_2020 | 0.0002 | 0.0254 | 0.0121 | <0.0001 | 0.0004 | 0.0001 | <0.0001 | 0.3342 | 0.2111 |
| embarked_2020 | 0.0001 | 0.0165 | 0.0062 | 0.0004 | <0.0001 | <0.0001 | <0.0001 | 0.2597 | 0.1709 |
| disembarked_2020 | 0.0003 | 0.0126 | 0.0043 | 0.0001 | <0.0001 | <0.0001 | <0.0001 | 0.2506 | 0.162 |
| Intransit_2020 | <0.0001 | 0.0255 | 0.0116 | <0.0001 | <0.0001 | <0.0001 | <0.0001 | 0.2838 | 0.1946 |
| covid19casesportugal_2020 | 0.6065 | 0.7945 | 0.9641 | 0.3342 | 0.2597 | 0.2506 | 0.2839 | <0.0001 | 0.0926 |
| covid19casesmadeira_2020 | 0.6182 | 0.8371 | 0.9277 | 0.2111 | 0.1709 | 0.162 | 0.1946 | 0.0926 | <0.0001 |

**Figure 4.** Significance Matrix—Monthly Data.

Furthermore, to study the indirect effect of the pandemic, a correlation significance test was conducted using worldwide COVID-19 case data. In the event of worldwide cases having a negative and significant relationship with the tourism variables, one can assume that the cases from the region itself are considered negligible since they represent approximately 0% of the global cases and deem the effect to be indirect due to the origin of these cases being outside of the autonomous region of Madeira. The reviewed literature regarding this relationship supported the idea that it should be positive and significant. However, the results show a positive correlation of worldwide COVID-19 cases with personnel costs and a negative correlation with the remaining variables, as displayed in Table 2, and their *p*-values reveal that these correlations are not statistically significant. The sample data could not perform a statistical inference to confirm the phenomenon described by [17,18], and thus was not in line with the idea of the existence of an indirect effect of the pandemic by having induced the fear of traveling.

**Table 2.** Correlation and Significance—World Cases.

| Variable | Correlation | *p*-Value |
|---|---|---|
| newguests_var | −0.1555 | 0.6293 |
| guests_2020 | −0.1564 | 0.6273 |
| sleepovers_202 | −0.1933 | 0.5473 |
| totalincome_2020 | −0.1571 | 0.6258 |
| totalroomincome_2020 | −0.1618 | 0.6155 |
| personalcosts_2020 | 0.1243 | 0.7003 |
| stopovers_2020 | −0.453 | 0.1391 |
| embarked_2020 | −0.5089 | 0.0911 |
| disembarked_2020 | −0.5225 | 0.0813 |
| intransit_2020 | −0.4745 | 0.1191 |
| worldcases | 1 | 0.0001 |
| newguests_var | −0.1555 | 0.6293 |

Focusing on municipal data, the correlation matrix of the municipalities represented in Figure 5 shows that the variable "COVID-19 cases in Madeira" has a combination of positive and negative relationships with the remaining variables related to tourism, represented in Figure 5, some of which have statistical significance, according to the results obtained in the *p*-value matrix in Figure 6.

| | newguests_month | cumulative_newguests | newguests_month_var | cumulative_newguests_var | guests_month | cumulative_guests | guests_month_var | cumulative_guests_var | sleepovers_month | cumulative_sleepovers | sleepovers_month_var | cumulative_sleepovers_var | averagestay | cumulative_averagestay | covidcases | populationaldensity | avgpopulation | longevity |
|---|---|---|---|---|---|---|---|---|---|---|---|---|---|---|---|---|---|---|
| newguests_month | 1 | 1 | -0.0088 | -0.0888 | 0.9999 | 0.9999 | -0.0638 | -0.0638 | 0.9922 | 0.999 | 0.0094 | 0.0094 | 0.4854 | 0.4854 | 0.802 | 0.8579 | 0.9089 | -0.3302 |
| cumulative_newguests | 1 | 1 | -0.0888 | -0.0888 | 0.9999 | 0.9999 | -0.0538 | -0.0638 | 0.9922 | 0.999 | 0.0004 | 0.0094 | 0.4854 | 0.4854 | 0.802 | 0.8579 | 0.9089 | -0.3302 |
| newguests_month_var | -0.0888 | -0.0888 | 1 | 1 | -0.0961 | -0.0961 | 0.999 | 0.999 | -0.1152 | -0.1149 | 0.9654 | 0.9654 | -0.1887 | -0.1887 | -0.2397 | -0.3597 | -0.3102 | 0.7144 |
| cumulative_rewguests_var | -0.0888 | -0.0888 | 1 | 1 | -0.0961 | -0.0961 | 0.999 | 0.999 | -0.1152 | -0.1149 | 0.9654 | 0.9654 | -0.1887 | -0.1887 | -0.2397 | -0.3597 | -0.3102 | 0.7144 |
| guests_month | 0.9999 | 0.9999 | -0.0961 | -0.0961 | 1 | 1 | -0.071 | -0.071 | 0.9931 | 0.9994 | 0.0025 | 0.0025 | 0.4884 | 0.4884 | 0.806 | 0.8627 | 0.9127 | -0.335 |
| cumulative_guests | 0.9999 | 0.9999 | -0.0961 | -0.0961 | 1 | 1 | -0.071 | -0.071 | 0.9931 | 0.9994 | 0.0025 | 0.0025 | 0.4884 | 0.4884 | 0.806 | 0.8627 | 0.9127 | -0.335 |
| guests_month_var | -0.0638 | -0.0638 | 0.999 | 0.999 | -0.071 | -0.071 | 1 | 1 | -0.0891 | -0.0897 | 0.9708 | 0.9708 | -0.1731 | 0.1731 | -0.2064 | -0.3305 | -0.2811 | 0.7035 |
| cumulative_guests_var | -0.0638 | -0.0638 | 0.999 | 0.999 | -0.071 | -0.071 | 1 | 1 | -0.0891 | -0.0997 | 0.9708 | 0.9708 | -0.1731 | -0.1731 | -0.2064 | -0.3305 | -0.2811 | 0.7035 |
| sleepovers_month | 0.9922 | 0.9922 | -0.1152 | -0.1152 | 0.9931 | 0.9931 | -0.0891 | -0.0891 | 1 | 0.9955 | -0.0091 | -0.0091 | 0.5128 | 0.5128 | 0.8148 | 0.8836 | 0.936 | -0.3294 |
| cumulative_sleepovers | 0.999 | 0.999 | -0.1149 | -0.1149 | 0.9994 | 0.9994 | -0.0897 | -0.0997 | 0.9955 | 1 | -0.0138 | -0.0138 | 0.5036 | 0.5036 | 0.817 | 0.876 | 0.9234 | -0.3458 |
| sleepovers_month_var | 0.0094 | 0.0004 | 0.9654 | 0.9654 | 0.0025 | 0.0025 | 0.9708 | 0.9708 | -0.0001 | -0.0138 | 1 | 1 | -0.0768 | -0.0768 | -0.1029 | -0.2395 | -0.1919 | 0.6482 |
| cumulative_sleepovers_var | 0.0094 | 0.0094 | 0.9654 | 0.9654 | 0.0025 | 0.0025 | 0.9708 | 0.9708 | -0.0001 | -0.0138 | 1 | 1 | -0.0768 | -0.0768 | -0.1029 | -0.2395 | -0.1919 | 0.6482 |
| averagestay | 0.4854 | 0.4854 | -0.1887 | -0.1887 | 0.4884 | 0.4884 | -0.1731 | -0.1731 | 0.5128 | 0.5036 | -0.0768 | -0.0768 | 1 | 1 | 0.6937 | 0.7001 | 0.6563 | -0.4867 |
| cumulative_averagestay | 0.4854 | 0.4854 | -0.1887 | -0.1887 | 0.4884 | 0.4884 | -0.1731 | -0.1731 | 0.5128 | 0.5036 | -0.0768 | -0.0768 | 1 | 1 | 0.6937 | 0.7001 | 0.6563 | -0.4867 |
| covidcases | 0.802 | 0.802 | -0.2397 | -0.2397 | 0.806 | 0.806 | -0.2064 | -0.2064 | 0.8148 | 0.817 | -0.1029 | -0.1029 | 0.6937 | 0.6937 | 1 | 0.9541 | 0.9095 | -0.5281 |
| populationaldensity | 0.8579 | 0.8579 | -0.3597 | -0.3597 | 0.8627 | 0.8527 | -0.3305 | -0.3305 | 0.8836 | 0.876 | -0.2395 | -0.2395 | 0.7001 | 0.7001 | 0.9541 | 1 | 0.985 | -0.615 |
| avgpopulation | 0.9089 | 0.9089 | -0.3102 | -0.3102 | 0.9127 | 0.9127 | -0.2811 | -0.2811 | 0.936 | 0.9234 | -0.1919 | -0.1919 | 0.6563 | 0.6563 | 0.9005 | 0.985 | 1 | -0.5675 |
| longevity | -0.3302 | -0.3302 | 0.7144 | 0.7144 | -0.335 | -0.335 | 0.7035 | 0.7035 | -0.3294 | -0.3458 | 0.6482 | 0.6482 | -0.4867 | -0.4867 | -0.5281 | -0.615 | -0.5675 | 1 |

**Figure 5.** Correlation Matrix—Municipality.

| | newguests_month | cumulative_newguests | newguests_month_var | cumulative_rewguests_var | guests_month | cumulative_guests | guests_month_var | cumulative_guests_var | sleepovers_month |
|---|---|---|---|---|---|---|---|---|---|
| newguests_month | <0.0001 | <0.0001 | 0.7951 | 0.7951 | <0.0001 | <0.0001 | 0.8523 | 0.8523 | <0.0001 |
| cumulative_newguests | <0.0001 | <0.0001 | 0.7951 | 0.7951 | <0.0001 | <0.0001 | 0.8523 | 0.8523 | <0.0001 |
| newguests_month_var | 0.7951 | 0.7951 | <0.0001 | <0.0001 | 0.7786 | 0.7786 | <0.0001 | <0.0001 | 0.7359 |
| cumulative_rewguests_var | 0.7951 | 0.7951 | <0.0001 | <0.0001 | 0.7786 | 0.7786 | <0.0001 | <0.0001 | 0.7359 |
| guests_month | <0.0001 | <0.0001 | 0.7786 | 0.7786 | <0.0001 | <0.0001 | 0.8356 | 0.8356 | <0.0001 |
| cumulative_guests | <0.0001 | <0.0001 | 0.7786 | 0.7786 | <0.0001 | <0.0001 | 0.8356 | 0.8356 | <0.0001 |
| guests_month_var | 0.8523 | 0.8523 | <0.0001 | <0.0001 | 0.8356 | 0.8356 | <0.0001 | <0.0001 | 0.7944 |
| cumulative_guests_var | 0.8523 | 0.8523 | <0.0001 | <0.0001 | 0.8356 | 0.8356 | <0.0001 | <0.0001 | 0.7944 |
| sleepovers_month | <0.0001 | <0.0001 | 0.7359 | 0.7359 | <0.0001 | <0.0001 | 0.7944 | 0.7944 | <0.0001 |
| cumulative_sleepovers | <0.0001 | <0.0001 | 0.7365 | 0.7365 | <0.0001 | <0.0001 | 0.7931 | 0.7931 | <0.0001 |
| sleepovers_month_var | 0.9782 | 0.9782 | <0.0001 | <0.0001 | 0.9942 | 0.9942 | <0.0001 | <0.0001 | 0.9789 |
| cumulative_sleepovers_var | 0.9782 | 0.9782 | <0.0001 | <0.0001 | 0.9942 | 0.9942 | <0.0001 | <0.0001 | 0.9789 |
| averagestay | 0.1302 | 0.1302 | 0.5784 | 0.5784 | 0.1275 | 0.1275 | 0.6108 | 0.6108 | 0.1067 |
| cumulative_averagestay | 0.1302 | 0.1302 | 0.5784 | 0.5784 | 0.1275 | 0.1275 | 0.6108 | 0.6108 | 0.1067 |
| covidcases | 0.003 | 0.003 | 0.4778 | 0.4778 | 0.0027 | 0.0027 | 0.5426 | 0.5426 | 0.0023 |
| populationaldensity | 0.0007 | 0.0007 | 0.2772 | 0.2772 | 0.0006 | 0.0006 | 0.3209 | 0.3209 | 0.0003 |
| avgpopulation | 0.0001 | 0.0001 | 0.3532 | 0.3532 | <0.0001 | <0.0001 | 0.4023 | 0.4023 | <0.0001 |
| longevity | 0.3213 | 0.3213 | 0.0135 | 0.0135 | 0.3139 | 0.3139 | 0.0157 | 0.0157 | 0.3226 |

**Figure 6.** Significance Matrix—Municipality.

The results obtained allow us to draw important conclusions regarding what was proposed in the research questions. They showed that the correlation between the same variables was significant when compared among municipalities. However, that did not happen when analyzed by month. This is observable in Figure 7, which compares the two subsets of data from 2019 to 2020 (before and during the pandemic, respectively) as a portrait of the pandemic's impact on the arrival of new guests in the region (see Figure 7).

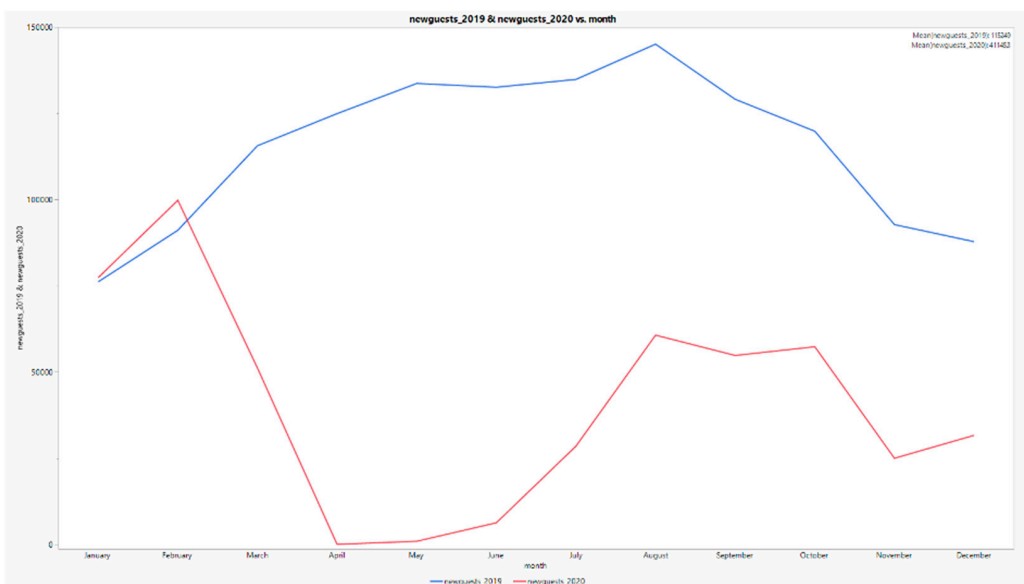

**Figure 7.** Evolution of new guests in lodging services in ARM in 2019 and 2020.

Furthermore, the data preparation phase provided a priori insight regarding population density. Some municipalities, such as Câmara de Lobos, one with a high population density, were highly impacted by the pandemic compared with most other municipalities. The results show that the variables "population density" and "average population" have shown a significant correlation with the variable "COVID cases", as seen and highlighted in Figure 8, leaning towards the existence of a statistically significant linear relationship between each of the two variables and the variation in COVID cases.

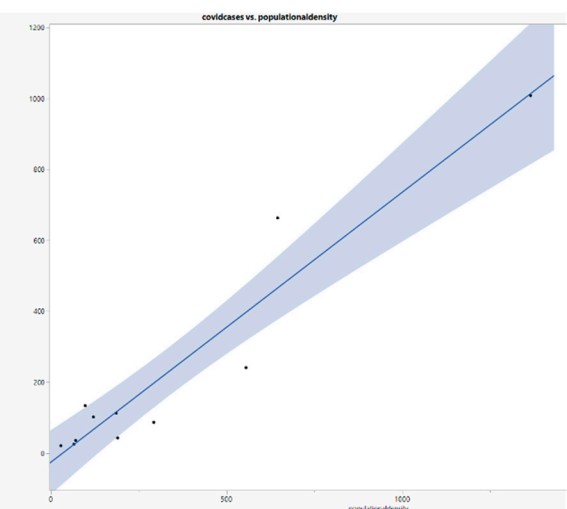

**Figure 8.** COVID-19 infection as a function of population density.

To complement the analysis of this relation, a clustering analysis was made using the relevant variables for this segment, which were "longevity", "cumulative new guests cumulative", "average stay", "COVID cases", "population density" and "average population".

Using the cubic clustering criteria, the optimal number of clusters was found to be 3. Then, the clustering method used was k-means, with the following results obtained in Table 3.

**Table 3.** K-means clustering results.

| Cluster | Longevity | Cumulative_Newguests | Averagestay | Covidcases | Populationaldensity | Avg Population |
|---------|-----------|----------------------|-------------|------------|---------------------|----------------|
| 1 | 51.2375 | 18,397.25 | 3.5752829 | 70 | 131.35 | 8913.375 |
| 2 | 41.65 | 30,880.5 | 5.1020125 | 428 | 600.55 | 39,358 |
| 3 | 42.4 | 28,4804 | 5.2462178 | 1008 | 1365.9 | 104,076.5 |

As we can observe, the results show that, groups of municipalities where the average stay was longer are associated with a higher incidence of COVID cases, and the same also goes for municipalities where the population density or average population was higher.

The municipalities and the three clusters they are inserted in are visually described in Figure 9 as a function of the number of COVID-19 infections and population density.

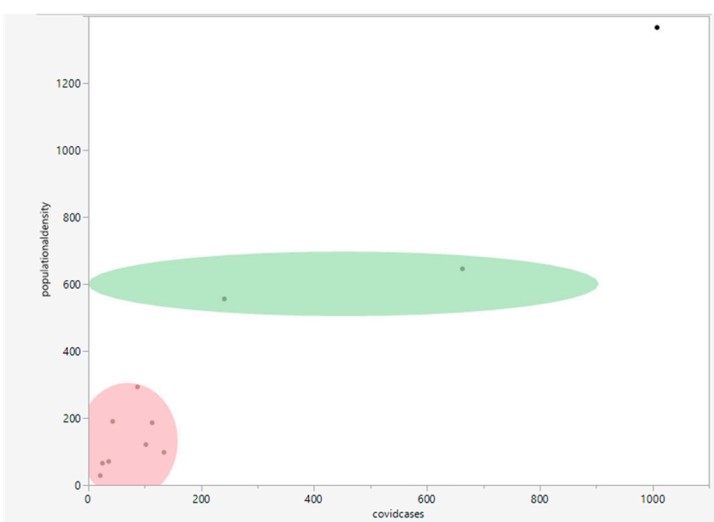

**Figure 9.** Cluster analysis representation, scrutinized by infection count and density.

Cluster 1 (Red): Calheta, Machico, Ponta do Sol, Porto Moniz, Ribeira Brava, Santana, São Vicente, and Porto Santo; the average number of COVID cases in this cluster was a low 70.13, associated with an average population density of 131.35 and an average population of 8913.

Cluster 2 (Green): Santa Cruz and Câmara de Lobos; as seen before, these are two of the most impacted regions, both of which have comparably high population densities. The cluster analysis shows that the average number of COVID cases in this cluster was 452, associated with an average population density of 600.55 and an average population of 8913.

Cluster 3 (Blue): Funchal; leading this cluster by itself, Funchal records a whopping 1008 cases, with a population density of 1366 and a population of 104,077.

The clustering results are a set of three clusters that are representative of three distinct tiers of population and population density levels. Cluster 1 is classified as having had a good performance in the context of the pandemic and represents the group of municipalities with a low population density mean and which have had fewer registered cases of infection by the virus. Cluster 2 is classified as having had an average performance in the context of the pandemic and represents the group of municipalities with an average population density that have registered an average number of infections. Lastly, Cluster 3 is classified as having had a bad performance in the context of the pandemic and represents the dense municipalities, which have registered the highest average number of infections. The results support the idea that we can predict how well an arbitrary municipality would do in a pandemic depending on its population density and that this variable is linearly related to COVID-19 cases.

The results obtained in this study concerning the impact of population density on the pandemic show a positive relationship between population density and the propagation of the coronavirus. This conclusion is aligned with previous studies [6–8,19–25,32]. These results show that municipalities with less dense clusters tend to perform well in controlling the propagation of the virus and its incidence. This finding corroborates what was found by [7] but contradicts what was obtained by [9]. This latter study found that low density cities had a higher impact on virus infections. However, it seems probable that this relationship is not as true to reality as the cities in the study—Singapore, Seoul, and Shanghai—classified as highly dense cities, have performed well in combating the coronavirus.

## 5. Conclusions

### 5.1. Research Contributions

In conclusion, the obtained results allowed us to reach conclusive findings and answer the four research questions. Most importantly, they provided insight into the impact of the COVID-19 Pandemic in the autonomous region of Madeira, the topic of this work.

Concerning the first research question, there is enough evidence to state that the COVID-19 Pandemic negatively impacted the tourism sector in the autonomous region of Madeira, corroborating the studies made by [13–16].

Regarding the second research question, the results were not conclusive enough to accept it. Interestingly, however, the highly ranked tourism destinations of Madeira and Porto Santo Islands still managed to have enough tourism activity to evade that indirect effect, which was significantly reflected in the results. However, due to the lack of statistical significance of the correlation of the variables that study the indirect impact on the region, such as "COVID-19 cases in Portugal", as well as "COVID-19 cases worldwide" and variables of the indicators of the tourism sector, there is not enough evidence to indicate that the unwinding of the pandemic throughout the world caused enough traveling fear and tourism constraints that could lead to an indirect negative impact on tourism in the region. These results go against previous studies [9,17].

Regarding the third research question, results showed that population levels and population density did indeed contribute positively to the transmission of the virus, with statistical significance associated with the corresponding correlations. This finding was also confirmed by the results obtained in the clustering analysis. The clustering analysis showed that high population-density municipalities had higher infection numbers. In conclusion, there is sufficient and conclusive evidence to conclude that there is a statistically significant linear relationship between population levels and density and COVID-19 cases. These allowed us to conclude that the population density of the municipalities significantly influences the spreading of the virus in the region.

Consequently, this fact reinforces the idea that higher population density and population levels, in general, helped spread the COVID-19 coronavirus. In comparison, lower population density and population levels tended to contain it. Conversely, the results do not corroborate the conclusions of other analyses that presumed that the density of the population was not at issue and could not be a determining factor in the proliferation of COVID-19 in general terms [9]. These results are also not in line with the results obtained in the early stages by [26], where this relationship was concluded not to be significant.

Lastly, the clustering analysis confirmed that there are patterns between the municipalities. The clustering analysis showed that it is possible to find groups of data that behave similarly among the municipalities, therefore answering the research question and setting the stage for future health crises in the autonomous region of Madeira, where the focus on preventive and restrictive measures should be on the clusters of municipalities where population density and registered COVID-19 cases are more prominent. Our research and analysis shall serve as documentation for future crises in the health sector by identifying and grouping municipalities with high risk associated with pandemics due to their population densities and for whom the regional government should target supplementary

measures to mitigate or possibly offset the effect of population density on the proliferation of a new virus.

*5.2. Managerial Implications*

The results obtained in this study have revealed patterns in the response of the autonomous region of Madeira to the COVID-19 pandemic that must be considered when facing future health crises, one of which is based on the municipalities' population density. Denser regions have been shown to have performed worse than less dense regions, and the highly contagious nature of the virus, especially in close proximity, means that in populationally denser regions, the proliferation of the virus was faster and in greater mass. This reality means that it is crucial that population density be identified as an indicator of risk in future health crises such as the one caused by the COVID-19 pandemic and that denser municipalities be targeted with heavier restrictions and sanitary measures since they have been shown to be more prone to perform worse in a pandemic situation. The study's findings thus reinforce the idea that the government's strategy and planning against pandemics should be guided by the differences in population parameters among the municipalities of the autonomous region of Madeira. Undoubtedly, the major consequences of the pandemic were the loss of lives, which is the main aspect the government should attempt to mitigate with these measures. However, it is also clear that the tourism sector benefits greatly from the government's effort to contain the virus in such a pandemic situation, and, in general terms, in the context of such a health crisis, the safer the region is considered to be, the more inviting to the tourism inflow it becomes, and these measures contribute to keeping regular rates of arrivals and tourist expenses, a major source of the domestic product, especially in Madeira.

*5.3. Limitations and Recommendations for Future Research*

While working on a topic related to tourism might induce one to pick a geographical area that is associated with large flows of tourism activity. Nonetheless, even though the autonomous region of Madeira is a highly regarded tourism destination internationally, the choice of this region for this study meant it became harder to obtain data from the sector and the local health authorities regarding the pandemic. As a comparison, on a larger geographical scale, equivalent data are public and available for access. Despite this not having been a major limiting factor in the making of this dissertation, it is advised that a pilot study and research on the availability of data be performed in order to acknowledge the resources and consider the possibility of expanding the scale of the project to a larger geographical area, possibly improving the process of collecting the resources needed for the goals aimed to be obtained with the project. Furthermore, the study was based on an a posteriori approach, evaluating the damages caused by the pandemic in the region and how the sector's indicators related to the propagation of the virus. So, the value of training a prediction model to estimate the development of the pandemic became redundant, although this is an interesting and useful approach suggested for future health crises. Additionally, this study only focused on the data until 2020. Therefore, future studies could go beyond this period and study the full 2021 year. Lastly, despite the results showing a correlation between the population density, the number of cases, and the number of tourists, the studied data does not allow for clarification as to whether the high number of tourists caused the high number of cases in high population-dense clusters, the higher population density caused the cases, or a combination of both. Therefore, future studies could explore other data sources to try to capture the cause behind the higher number of cases.

**Author Contributions:** Conceptualization, J.N. and P.R.; methodology, J.N., P.R. and N.A.; software, J.N.; validation, P.R. and N.A.; investigation, J.N. and P.R.; resources, J.N.; data curation, J.N.; writing—original draft preparation, J.N.; writing—review and editing, J.N. and P.R.; visualization, J.N. All authors have read and agreed to the published version of the manuscript.

**Funding:** This work was supported by national funds through FCT (Fundação para a Ciência e a Tecnologia), under the project—UIDB/04152/2020—Centro de Investigação em Gestão de Informação (MagIC)/NOVA IMS.

**Institutional Review Board Statement:** Not applicable.

**Informed Consent Statement:** Not applicable.

**Data Availability Statement:** Not applicable.

**Conflicts of Interest:** The authors declare no conflict of interest.

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
