# Peer review of "The Impact of the COVID-19 Pandemic on the Tourism Sector in the Autonomous Region of Madeira"

_sustainability, doi:10.3390/su151612298_

Round 1

Reviewer 1 Report

 Dear Authors,    

The article titled "The Impact of the Covid-19 Pandemic on the Tourism Sector in the Autonomous Region of Madeira" aims to assess the impact of the pandemic on the tourism sector in the Autonomous Region of Madeira by examining data from the sector through its indicators and examining their relationship to COVID-19 data.  

After reading the article, I have the following comments and suggestions for improving it:   

 Abstract: Summary : should be revised according to the journal's guidelines. There is a lack of information about what methods were used to conduct the study and what are the results of the study?  

In the Introduction 

The introduction to the topic is interesting and based on the literature.  However, the paper poses too many research questions. I suggest putting them in order for better reception. 

Similarly, the purpose of the paper is presented twice.

The first is very long and makes it incomprehensible: 

"This study aims to assess the impact of the pandemic on the tourism sector in the Autonomous Region of Madeira by examining the sector's data through its indicators and exploring their relationship with COVID-19 data, as well as delving into the collected data by using Data Mining tools to fill the aforementioned research gaps. in particular, by adding contributions to the study of the controversial link between population density and pandemics and using log data to explore the indirect impact of pandemics on tourism, while adding to the results obtained from searching, analyzing and transforming the collected data into knowledge that can answer the research questions and can draw meaningful conclusions in the context of the study topic.

The second objective echoes the first:

"In summary, this study aims to assess the extent of the direct and indirect impact of the pandemic on the tourism sector, answering the first and second research questions, respectively. In addition, to answer the third research question, the goal is to analyze the impact of population density on the spread of coronavirus in the Autonomous Region of Madeira. Finally, to answer the fourth research question, the study aims to determine whether there are similarities between groups of municipalities and what the differences between clusters might mean in terms of management conclusions."

2 Literature review and research hypotheses 

The theoretical part is well presented based on global literature.  

I do not understand why the hypotheses of the work were placed in this part? 

[290-293] - In my opinion, I think there are too many research hypotheses.  

3.Materials and methods 

Information about the research area was missing. Where was the research conducted?  Why is this region important for tourism?

The Results  

The results were presented and described in a good way, they are very interesting and important for the development of tourism. However, I would suggest doing some organizing of the obtained results in the form of tables. The text needs to be clear and easy to read. 

All in all, I recommend this paper for publication in the Journal “Sustainability” after minor changes. 

Kind regards, 

Reviewer

Author Response

Reviewer 1

Dear Authors,    

The article titled "The Impact of the Covid-19 Pandemic on the Tourism Sector in the Autonomous Region of Madeira" aims to assess the impact of the pandemic on the tourism sector in the Autonomous Region of Madeira by examining data from the sector through its indicators and examining their relationship to COVID-19 data. 

After reading the article, I have the following comments and suggestions for improving it:   

 Abstract: Summary: should be revised according to the journal's guidelines. There is a lack of information about what methods were used to conduct the study and what are the results of the study?  

Answer (A): Thank you for highlighting this situation. We have extended the abstract to be clearer in the methods employed. However, we consider that results are already mentioned in the last sentence of the abstract.

In the Introduction 

The introduction to the topic is interesting and based on the literature.  However, the paper poses too many research questions. I suggest putting them in order for better reception. 

Similarly, the purpose of the paper is presented twice.

The first is very long and makes it incomprehensible: 

"This study aims to assess the impact of the pandemic on the tourism sector in the Autonomous Region of Madeira by examining the sector's data through its indicators and exploring their relationship with COVID-19 data, as well as delving into the collected data by using Data Mining tools to fill the aforementioned research gaps. in particular, by adding contributions to the study of the controversial link between population density and pandemics and using log data to explore the indirect impact of pandemics on tourism, while adding to the results obtained from searching, analyzing and transforming the collected data into knowledge that can answer the research questions and can draw meaningful conclusions in the context of the study topic.

The second objective echoes the first:

"In summary, this study aims to assess the extent of the direct and indirect impact of the pandemic on the tourism sector, answering the first and second research questions, respectively. In addition, to answer the third research question, the goal is to analyze the impact of population density on the spread of coronavirus in the Autonomous Region of Madeira. Finally, to answer the fourth research question, the study aims to determine whether there are similarities between groups of municipalities and what the differences between clusters might mean in terms of management conclusions."

A: Indeed, these introductory paragraphs were too long and somewhat repetitive. Thank you for calling our attention to this fact. We have now revised them.

2 Literature review and research hypotheses 

The theoretical part is well presented based on global literature.  

I do not understand why the hypotheses of the work were placed in this part? 

[290-293] - In my opinion, I think there are too many research hypotheses.  

A: We value your opinion and the one put forward by another reviewer. Therefore, we removed the hypotheses and focused on finding direct answers to the research questions.

3.Materials and methods 

Information about the research area was missing. Where was the research conducted?  Why is this region important for tourism?

A: Once again, we thank the reviewer for calling our attention to clarification. We have added a sentence in the Introduction to explain why we focused on the Madeira islands and where they are.

The Results  

The results were presented and described in a good way, they are very interesting and important for the development of tourism. However, I would suggest doing some organizing of the obtained results in the form of tables. The text needs to be clear and easy to read. 

A: We have revised the Results and discussion chapter to make it clear and easier to read.

All in all, I recommend this paper for publication in the Journal “Sustainability” after minor changes. 

A: Thank you very much for your comments and constructive recommendations.

Reviewer 2 Report

This paper proposes an original study that deals with a very interesting topic. Although this article is original, this reviewer suggests a revision.

Section Introduction: the introduction should be revised. It needs a clean-up, is too long. Some of the information should be replaced in the Literature Review. The authors should be clearer and more direct. At the moment, the front end of the paper is quite vague and one does not get a sense that this kind of study is really needed. A better way to begin this would be to talk about the problem in the first part (explain why this research is relevant), give a brief overview of past studies in the second, show what you are doing different in the third, and use the fourth for the main goals and novelty of the study.

The authors should consider including the Literature Review section, relevant studies on the Covid-19 impact I Europe and specially in Portugal. Several studies on other countries are mentioned, however there are so many relevant studies on the pandemic impact in the tourism Portuguese industry that were neglected. The hypothesis’ development is completely wrong. Better saying the hypothesis were not developed and supported at all. As they are, they are only research questions. Also, the several hypothesis are not related and it is very difficult to understand how they can all be together in the same study and conceptual model.

Section Conclusion: the results are very interesting hence the conclusions and discussion are extremely poor. The authors should include the most relevant implications for theory and practice. This author do not understand the implications of this study to science.

Some typos and grammar failures should be corrected. Also, some relevant literature on the COVID-19 impact in tourism is not referenced.

Author Response

Reviewer 2

This paper proposes an original study that deals with a very interesting topic. Although this article is original, this reviewer suggests a revision.

Section Introduction: the introduction should be revised. It needs a clean-up, is too long. Some of the information should be replaced in the Literature Review. The authors should be clearer and more direct. At the moment, the front end of the paper is quite vague and one does not get a sense that this kind of study is really needed. A better way to begin this would be to talk about the problem in the first part (explain why this research is relevant), give a brief overview of past studies in the second, show what you are doing different in the third, and use the fourth for the main goals and novelty of the study. 

The authors should consider including the Literature Review section, relevant studies on the Covid-19 impact I Europe and specially in Portugal. Several studies on other countries are mentioned, however there are so many relevant studies on the pandemic impact in the tourism Portuguese industry that were neglected. The hypothesis’ development is completely wrong. Better saying the hypothesis were not developed and supported at all. As they are, they are only research questions. Also, the several hypothesis are not related and it is very difficult to understand how they can all be together in the same study and conceptual model.

Section Conclusion: the results are very interesting hence the conclusions and discussion are extremely poor. The authors should include the most relevant implications for theory and practice. This author do not understand the implications of this study to science. 

A: Conclusions were revised to address the reviewer's concerns. We hope the section is now aligned with what the reviewer expected.

Some typos and grammar failures should be corrected. Also, some relevant literature on the COVID-19 impact in tourism is not referenced. 

A: Full proofreading was done after all the implemented changes. We thank the reviewer for the valuable comments and suggestions.

Reviewer 3 Report

Dear Authors,

The article deals with a very topical issue in the tourism sector of a well-known destination. The research procedure used may be useful in the study of other destinations. Below are some notes to think about and implement in order to better understand the contents of the manuscript.

1. lines 83-108 formulate the research tasks, but are essentially research hypotheses, which again appear in section 2.2 Research Hypotheses. Please think about combining these thoughts and rewording them in one section of the manuscript - perhaps in section 2.2.

2. please explain the abbreviations: DREM, RAM, APRAM, SESARAM.

3. there is a missing citation of fig 2 in the text of the manuscript.

4. the explanations of the axes in figs 1, 2, 8 are unreadable.

Author Response

Reviewer 3

The article deals with a very topical issue in the tourism sector of a well-known destination. The research procedure used may be useful in the study of other destinations. Below are some notes to think about and implement in order to better understand the contents of the manuscript.

  1. lines 83-108 formulate the research tasks, but are essentially research hypotheses, which again appear in section 2.2 Research Hypotheses. Please think about combining these thoughts and rewording them in one section of the manuscript - perhaps in section 2.2.
  2. please explain the abbreviations: DREM, RAM, APRAM, SESARAM. 
  3. there is a missing citation of fig 2 in the text of the manuscript.
  4. the explanations of the axes in figs 1, 2, 8 are unreadable.

A: Dear reviewer, thank you for your comments. We have addressed all the pointed out issues.

Reviewer 4 Report

Please see the document attached.

Only minor revisions of the English language are needed. 

Author Response

Reviewer 4

The paper ‘The Impact of the Covid-19 Pandemic on the Tourism Sector in the Autonomous Region of Madeira’ is in line with the topic of the special issue of Sustainability. Overall, I see this paper as having merit. The research hypotheses are clearly presented, the methodology is sound and detailed, the discussion is balanced.

However, a key issue of the paper is related to the link between population density and tourism impact. I am not saying that there is no such connection, as several research focusing on this topic attest (Jeon & Yang, 2021; Schumde et.al., 2021 etc. - see reference suggestions below), but the current paper presents the two elements –pandemic impact on tourism industry and weight of population density in the propagation of the virus independently and in this case, I do not see the reason for including variables such as population density, longevity, average population (Table 2, page 9-10) when analysing the impact on tourism.

  • The authors should expand on the interdependencies between high/low population density areas and tourism demand.

A: We made several changes in the introduction and the literature review. We think it is now clear. Thank you for highlighting this point.

  • How do strategic tourism products for Madeira, namely nature tourism and sun and sea relate to tourism demand during the analysed period? How do the municipalities in the three clusters (page 15-16) relate to the main tourism destination within the study area? There is only one line (560) making reference to this point.

A: Thank you. We added additional information about Madeira Island and the tourism impact on it.

More information should also be provided for a thorough overview of the situation in Madeira, which forms the current study area. The authors provide a short background depiction of the area (lines 41- 54), but further information would also be needed in order to get the whole picture, namely:

  • Why was only one year (2020) chosen to analyse the impact? Did the situation in 2021 change completely?

A: When the study was concluded, there was only data for 2020. We have now included this as a limitation of the study.

  • How restrictive were government measures regarding the flow of people? Was there a lockdown (since/ until)? What were the government measures to support tourism sector? The only mention is found on page 12 (lines 461-466)

A: We consider your point interesting but exploring that could generate other avenues of research that could be outside our research scope.

  • Were the effects of pandemic distributed homogenously throughout the country? How does Madeira compare to other tourism destinations in Portugal?

Useful papers on this topic might be:
Costa, C. (2021). The impact of the COVID-19 outbreak on the tourism and travel sectors in Portugal: Recommendations for maximising the contribution of the European Regional Development Fund (ERDF) and the Cohesion Fund (CF) to the recovery. DG REGIO, European Commission.
Rebola, F., Loures, L., Ferreira, P., & Loures, A. (2022). Inland or Coastal: That’s the Question! Different Impacts of COVID-19 on the Tourism Sector in Portugal. Sustainability, 14(23), 16248. SANTOS, Norberto; MOREIRA, Claudete Oliveira. Uncertainty and expectations in Portugal's tourism activities. Impacts of COVID-19. Research in Globalization, 2021, 3: 100071.
Yang, T. T., Ruan, W. Q., Zhang, S. N., & Li, Y. Q. (2021). The influence of the COVID-19 pandemic on tourism demand for destinations: an analysis of spatial heterogeneity from a multi-scale perspective. Asia Pacific Journal of Tourism Research, 26(7), 793-810.

Round 2

Reviewer 2 Report

The authors made a significant effort to address the reviewers' concerns.

Author Response

Sustainability Reviewers Comments

Comments and Suggestions for Authors

We appreciate all reviewers’ kind words, comments, and suggestions. Your comments were vital in our manuscript improvement.

Regarding the reviewer 4 comments and questions, we thank you once more the reviewer for the valuable comments and suggestions on improving and strengthening our manuscript.

As for the two questions the reviewer considers are still to be addressed regarding the connection between tourism and density, namely:

- what are the interdependencies between high/low population density areas and tourism demand?

- How do the municipalities in the three clusters (pages 15- 16) relate to the main tourism destination within the study area

We consider these questions to be answered in the literature review (lines 223 – 252). Several studies have demonstrated the impact of the COVID-19 pandemic on tourism. However, not all studies agree that high population density is highly correlated to COVID-19 cases. Therefore, this study explores this subject in order to understand the relationship between the pandemic severity and population density. As mentioned in lines 481 to 519, including in Figure 10, the impact of the pandemic was highly correlated with the density of the municipalities and the number of guests. From the studied data, it is not possible to assess if the high number of tourists caused the high number of cases in high population-dense clusters, the higher population density, or a combination of both. We have considered your comments and added this as a limitation and an avenue for future research to explore.

Reviewer 4 Report

I carefully read the revised version of the paper, and although you made a lot of changes, for which I congratulate you, I feel that the first and most important issue -  the link between population density and tourism impact, has not been addressed. Indeed, you mentioned that 'There appears to be a controversy in results regarding whether population density has a negative impact on the propagation and severity of pandemics' and further detail this issue (lines 46-59), but again, it does not make any reference to the connection between population density and tourism. So, I feel I must ask again:

- what are the interdependencies between high/low population density areas and tourism demand?

- How do the municipalities in the three clusters (page 15-16) relate to the main tourism destination within the study area

Author Response

(The authors gave the same response as above.)
